# Epigenetic regulation of defense genes by histone deacetylase1 in human cell line-derived macrophages promotes intracellular survival of *Leishmania donovani*

**Gargi Roy[1], Harsimran Kaur Brar[1], Rohini Muthuswami[2]\*, Rentala Madhubala[1]\***

**1** MolecularParasitology Laboratory, School of Life Sciences, Jawaharlal Nehru University, New Delhi, India,
**2** Chromatin Remodeling Laboratory, School of Life Sciences, Jawaharlal Nehru University, New Delhi, India

\* rentala@outlook.com (RMu); rohinim@mail.jnu.ac.in (RMa)

**Data Availability Statement:** All relevant data are within the manuscript and its Supporting Information files.

## Abstract

*Leishmania donovani*, an intracellular protozoan parasite upon infection, encounters a range of antimicrobial factors within the host cells. Consequently, the parasite has evolved mechanisms to evade this hostile defense system through inhibition of macrophage activation that, in turn, enables parasite replication and survival. There is growing evidence that epigenetic down-regulation of the host genome by intracellular pathogens leads to acute infection. Epigenetic modification is mediated by chromatin remodeling, histone modifications, or DNA methylation. Histone deacetylases (HDACs) removes acetyl groups from lysine residues on histones, thereby leading to chromatin remodeling and gene silencing. Here, using *L. donovani* infected macrophages differentiated from THP-1 human monocytic cells, we report a link between host chromatin modifications, transcription of defense genes and intracellular infection with *L. donovani*. Infection with *L. donovani* led to the silencing of host defense gene expression. Histone deacetylase 1 (HDAC1) transcript levels, protein expression, and enzyme activity showed a significant increase upon infection. HDAC1 occupancy at the promoters of the defense genes significantly increased upon infection, which in turn resulted in decreased histone H3 acetylation in infected cells, resulting in the down-regulation of mRNA expression of host defense genes. Small molecule mediated inhibition and siRNA mediated down-regulation of *HDAC1* increased the expression levels of host defense genes. Interestingly, in this study, we demonstrate that the silencing of HDAC1 by both siRNA and pharmacological inhibitors resulted in decreased intracellular parasite survival. The present data not only demonstrate that up-regulation of *HDAC1* and epigenetic silencing of host cell defense genes is essential for *L. donovani* infection but also provides novel therapeutic strategies against leishmaniasis.

## Author summary

Visceral leishmaniasis is a neglected tropical disease caused by the protozoan parasite *Leishmania donovani*. These intracellular parasites replicate inside phagolysosomes of the

**Funding:** R.M. was funded by EMR/2016/004948 from Science and Engineering Research Board, India (https://www.serbonline.in/SERB/HomePage. do) and VI-D&P/569/2016-17/TDT/C from Department of Science and Technology, India (www.dst.gov.in). GR was supported by D.S. Kothari Fellowship and HKB was supported by fellowship from CSIR. The funders had no role in study design, data collection and analysis, decision to publish, or preparation of the manuscript.

**Competing interests:** The authors have declared that no competing interests exist.

macrophages and have evolved mechanisms that allow survival within the hostile environment of their hosts. They have evolved strategies to dramatically modify the transcriptome and proteome content of the host cells they infect, facilitating their survival within the host cell. Here, we have defined a mechanism by which *Leishmania donovani* subverts host cell defense genes by manipulating the epigenetic control of defense gene expression. The intracellular protozoan parasite, *Leishmania*, results in gene silencing of innate host defense genes. The underlying molecular mechanisms assisting such modifications were inferred from the results. An increase in the transcript, protein levels and activity of host HDAC1 was observed at 6 h post-infection. This data confirms the role of HDAC1 in the silencing of the host defense genes and enabling the intracellular survival of the parasite. Our results decode a new insight into epigenetic regulation of host cell defense mechanisms. These findings will facilitate the identification of novel strategies and will promise newer ways to control parasite growth.

## Introduction

Visceral leishmaniasis (VL), also called kala-azar, is a widespread and highly endemic disease. It is caused by an intracellular protozoan parasite, *Leishmania donovani*. Visceral leishmaniasis continues to be a major public health concern, especially in Africa, Asia, and South America [1, 2]. *Leishmania* parasites have a digenetic life cycle that alternates between flagellated promastigote form that lives in the gut of the sand fly and the amastigote form that replicates in the macrophages of its mammalian host. *Leishmania* has evolved to evade the host epigenome thus, enabling parasite replication and survival [3, 4]

Once inside the mammalian cells, the parasites, encounter a range of antimicrobial factors such as defensins, before binding and engulfment by host macrophages. Antimicrobial peptides (AMPs) are components of the innate immune system [5, 6] and have potent antimicrobicidal activity against prokaryotic and eukaryotic pathogens as well as viruses. Numerous studies have reported differential expression of defensins and other antimicrobial peptides upon pathogenic and parasitic infection [6–10]. Members of the alpha- and theta-defensins, magainins, and cathelicidins have been shown to have substantially higher leishmanicidal activity [11]. Neutrophil extracellular trap (NET) proteins ensnare and kill microorganisms are dependent on myeloperoxidase (MPO).

Furthermore, inhibition of MPO affects NET levels negatively [12]. Leptin (LEP) deficiency has been shown to facilitate VL pathogenesis. Up-regulation of IL-1α, IL-1β, IL-8, TNF-α, IFN-γ, IL-12, and IL-2 in *L. donovani* infected peripheral blood mononuclear cells (PBMCs) has been reported previously [13].

Several independent studies have demonstrated that various pathogens like viruses and bacteria re-modulate host epigenetics for their survival as well as infection within the host [3, 4, 14]. Cell reprogramming involves epigenetic changes by chromatin remodeling, histone modifications, and DNA methylation for normal development and maintenance of cellular differentiation [3, 15]. Histone deacetylases (HDACs) remove acetyl groups from lysine residues on histones, thereby leading to chromatin remodeling and gene silencing [14, 16, 17]. They are balanced by the activities of their counterparts, i.e., histone acetyltransferases. Small molecules that inhibit HDAC function have gained growing attention as potential drug targets in the last ten years as the role of aberrant epigenetic alteration in addition to genetic mutations became more evident in various diseases. HDAC inhibitors are being investigated as drugs for a wide

range of diseases, including cancers and infectious diseases such as HIV/AIDS, and several parasitic diseases [18].

In the present study, we investigated if *Leishmania* infection results in the reprogramming of the host epigenome. Therefore, towards this end, we investigated the expression of host HDAC1 and its role in chromatin modulation of host defense genes and parasite survival post-infection of macrophages with *L.donovani*. We report that the down-regulation of key host defense genes is critical for the intracellular survival of *L. donovani*. Our data confirm the role of HDAC1 in the silencing of the host defense genes and enabling the intracellular survival of the parasite. Considering the toxicity of drugs used in chemotherapy of leishmaniasis and the unavailability of the appropriate vaccine, a search is on to find newer targets to suppress parasite growth [19]. In light of this, our findings not only demonstrate the epigenetic mechanism of host cell subversion but also provide a novel therapeutic strategy for treating visceral leishmaniasis.

## Materials and methods

### Materials

Acetylated-Lysine rabbit monoclonal antibody (Catalog No.Ac-K$^2$-100, 9814), HDAC1 mouse monoclonal antibody (Catalog No 10E2, 5356), Histone H3 mouse monoclonal antibody (Catalog No 1B1B2, 14269), Anti-mouse IgG, HRP-linked antibody (Catalog No 7076), and Anti-rabbit IgG, HRP-linked antibody (Catalog No7074S) were purchased from Cell Signaling Technology, USA.

### Parasite and mammalian cell culture conditions

*L. donovani* Bob (LdBob/strain/MHOM/SD/62/1SCL2D) [20, 21] initially acquired from Dr. Stephen Beverly (Washington University, St. Louis, MO) were used in this study. *L. donovani* were maintained at 22˚C in M199 medium (Sigma-Aldrich, USA) supplemented with 100 units/ml penicillin (Sigma-Aldrich, USA), 100 µg/ml streptomycin (Sigma-Aldrich, USA) and 10% heat inactivated fetal bovine serum (FBS) (Biowest, UK).

THP-1 cells, an acute monocytic leukemia-derived human cell line (202 TIB; American Type Culture Collection, Rockville, MD) were grown in RPMI medium (Sigma-Aldrich, USA) supplemented with 10% heat inactivated FBS (Biowest, UK), 100 U/ml each of penicillin and 100 mg/ml streptomycin (Sigma-Aldrich, USA) and maintained at 37˚C with 5% $CO_2$.

### Macrophage infection

THP-1 cells ($10^5$ cells/ml) were treated with phorbol-12-myristate-13-acetate (PMA) (Sigma-Aldrich, USA) (50 ng/ml for 48 h) to induce differentiation into macrophage like cells before infection. PMA treated cells were incubated with *L. donovani* for 3 h at a multiplicity of infection (MOI) of 20 parasites per cell. Infected cells were then harvested at different time points (6, 12, and 24 h). Infections were confirmed using Giemsa (Sigma-Aldrich, USA) and Propidium Iodide (PI) (Sigma-Aldrich, USA) staining. For suppression of endogenous HDAC1 expression, cells were treated with sodium butyrate (NaB) (Sigma-Aldrich, USA) or Vorinostat ($C_{14}H_{20}N_2O_3$) (Sigma-Aldrich, USA) at concentrations 1.25–20 mM and 1–5 µM, respectively. The inhibitors were suspended in their respective solvents as per the manufacturer's instructions.

## RNA extraction and quantitative RT-PCR (qRT-PCR)

Total RNA from the infected macrophages was isolated using the TRIZOL reagent (Sigma-Aldrich, USA). RNA was precipitated by phenol-chloroform treatment and dissolved in DEPC-treated RNase free water and quantified by spectrophotometric analysis, cDNA was prepared from 4 **μg** of total RNA using First Strand cDNA Synthesis Kit as per manufacturer's instructions (Thermo Fisher Scientific, USA) using random hexamer primers. The subsequent cDNA was analysed by quantitative real-time (qRT-PCR) experiments (Applied Biosystems, 7500 Fast Real-Time PCR System, CA, USA) using SYBR Green PCR Master Mix (Thermo Fisher Scientific) and primers for the respective genes (S1 Table). The following reactions conditions were used: 50˚C for 2 min, with an initial activation step at 95ºC for 10 min, followed by 40 cycles at 95˚C for 30 s, 62˚C for 1 min. *RNU6A* was used as the housekeeping gene for normalization. The basal level of transcript expression in uninfected cells was used for data normalization and to assess relative abundance. The comparative threshold cycle method was used to quantify the change of gene expression for relative quantification ($2^{\wedge -\Delta\Delta CT}$), as reported earlier [22].

## Chromatin immunoprecipitation (ChIP)

ChIP analysis was performed with chromatin of *L. donovani* infected and uninfected THP-1 cells ($10^5$ cells/ml) followed by qRT-PCR using promoter specific primers (S1 Table) as reported previously [23, 24]. Briefly, the uninfected and infected THP-1 cells were cross-linked with formaldehyde (final concentration 1%) followed by quenching of the reaction by glycine (final concentration 125 mM). Subsequently, the cells were washed with ice cold PBS and lysed in a buffer containing 50 mM HEPES-KOH (pH 7.5), 140 mM NaCl, 1 mM EDTA, 1% Triton X-100, 0.1% sodium deoxycholate and 0.1% SDS and protease inhibitors. Chromatin was sheared by sonication (37 times for 30 sec; Ultrasonic water bath, MRC Ltd, Israel) to obtain 200 to 600 bp chromatin fragments. Histone and HDAC1 bound DNA was immunoprecipitated overnight at 4˚C using Ac-H3 (2.5 μg/25 μg chromatin extract) and HDAC1 (2.5 μg/ 25 μg chromatin extract) antibodies. Subsequently, the complex was pulled down using protein G beads pre-adsorbed in 75 ng/μl sonicated salmon sperm DNA and 0.1 μg/μl BSA. The respective chromatin extracts were also incubated with Protein G beads alone to serve as negative controls. The immune complexes were eluted from the beads by washing (three times) in a buffer containing 0.1% w/v sodium dodecyl sulphate, 1% Triton X-100, 2 mM EDTA, 150 mM NaCl, and 20 mM Tris-Cl (pH 8.0), followed by a final washing in a second buffer containing 0.1% sodium dodecyl sulphate, 1% Triton X-100, 2 mM EDTA, 500 mM NaCl, and 20 mM Tris-Cl (pH 8.0). The cross-links of the eluted protein-DNA complexes were reversed using RNase (12.5 μg/25 μg chromatin extract) and Proteinase K (125 ng/25 μg chromatin extract). The immunoprecipitated DNA fragments were eluted in 1% sodium dodecyl sulphate and 100 mM NaHCO$_3$ and purified using phenol: chloroform. The eluted DNA was used for qRT-PCR using primers spanning -117 to +116 of *MPO*, -131 to +91 of *HAMP*, -47 to +166 of *GNLY*, -135 to +75 of *DEFA1*, -89 to +129 of *DEFA4*, -49 to +161 of *DEFA5*, -111 to +119 of *DEFA6*, -125 to +92 of *DEFB1*, -69 to +111 of *DEFB4*, -97 to +103 of *PTEN* and -125 to +59 of *IL8* of their respective promoter regions (S1 Table). Fold change difference of the C$_T$ values obtained for the negative antibody control, and uninfected chromatin extract control was used to calculate the relative enrichment of each DNA fragment.

## Cell fractionation

Nuclear extracts from *L. donovani* infected and uninfected THP-1 cells ($10^5$ cells/ml) were prepared using a protocol by Rockland antibodies and assays (https://rockland-inc.com/Nuclear-

Extract-Protocol.aspx). The cells harvested at different time points were washed gently with PBS buffer and centrifuged at 1000 rpm for 1 min. The pellets were resuspended in 5 pellet volume of cytoplasmic extract (CE) buffer containing 10 mM HEPES-KOH, pH 7.6, 60 mM KCl, 1mM EDTA, 0.075% NP40, 1 mM DTT and 1 mM PMSF for isolation of cytoplasmic extract. After incubation on ice for 3 min, samples were centrifuged at 1000 to 1500 rpm for 4 min. The cytoplasmic extract was collected and stored. The nuclear pellet was washed with a CE buffer without detergent and centrifuged at 1000 to 1500 rpm for 4 min. One pellet volume nuclear extract (NE) buffer containing 20 mM Tris-Cl, pH.8.0 1.5 mM MgCl$_2$, 420 mM NaCl, 0.2 mM EDTA and 25% glycerol was used to resuspend nuclear pellet. The salt concentration was adjusted to 400 mM using 5 M NaCl. An additional pellet volume of NE buffer was further added. The nuclear pellets in NE buffer were incubated on ice for 10 min with periodic vortexing. The samples were then centrifuged at maximum speed for 10 min to pellet any irrelevant cell debris and the extracts were stored at -80˚C with 20% glycerol.

## Immunoblotting

HDAC1 protein expression in the nuclear fraction of uninfected and infected cells was analyzed by immunoblotting. Approximately 5 μg of nuclear protein fraction was electrophoresed on a 10% SDS-PAGE gel followed by transfer to nitrocellulose membrane using an electrophoretic transfer cell (Bio-Rad Laboratories, USA). After blocking with 5% skimmed milk, the membrane was further incubated with HDAC1 mouse monoclonal antibody (1:1000) previously histone H3 mouse monoclonal antibody (1:1000). The membrane was then washed with Tris-buffered saline containing 0.05% Tween 20 and incubated with horseradish peroxidase (HRP)-conjugated anti-mouse IgG antibody (1:5000). HDAC1 expression levels were normalized with that of H3 and quantitated by densitometry using ImageJ software. The separation of nuclear from the cytoplasmic extract was confirmed by using antibody against Histone 3 for nuclear fraction (Fig 3C).

## Immunoprecipitation

Uninfected THP-1 cells and *L. donovani* infected cells were harvested 6 and 24 h post infection, washed with ice-cold phosphate-buffered saline and lysed in lysis buffer containing 50 mM Tris-HCl (pH 7.4), 150 mM NaCl, and 1% Triton X-100 (vol/vol), supplemented with Complete protease inhibitor cocktail (Roche Diagnostics, Mannheim, Germany) as described previously [25]. Sample lysates were incubated with anti-HDAC1 (2.5 μg/25 μg chromatin extract) antibody overnight at 4˚C. Subsequently, the complex was pulled down using protein G beads pre-adsorbed on 75 ng/μl salmon sperm DNA and BSA (0.1 μg/μl). The protein lysates without incubation with anti-HDAC1 antibody were also incubated with Protein G beads to serve as negative controls. The immunoprecipitated samples were washed three times with the lysis buffer and finally eluted with 50 mM Tris-HCl (pH 6.8) and 1% SDS. Immunoprecipitated samples were confirmed by immunoblotting of the eluates with HDAC1specific antibody.

## HDAC activity assay

The nuclear fractions of infected and uninfected macrophages were harvested as described earlier. They were used to perform the HDAC protein activity assay using Epiquik HDAC Activity Assay Kit (Fluorometric) (Epigentek, USA). The experiment was conducted as per the manufacturer's instructions. Briefly, nuclear extracts (10 μg) were incubated with biotinylated HDAC substrate for 60 min at 37˚C followed by incubation with capture antibody for 60 min

and detection antibody for 30 min. Subsequently, the samples were incubated with Fluoro-Developer for 5 min at room temperature.

Furthermore, for the analysis of the activity of HDAC1, specifically in the reaction samples, 1 ug of immunoprecipitated HDAC1 protein obtained from infected and uninfected macrophages were used for the assay.

The amount of deacetylated histone, which is proportional to HDAC enzyme activity, was captured fluorometrically using Varioskan Flash Multiplate Reader (Thermo Fisher Scientific, USA) at 530 nm excitation and 590 nm emission. The enzymatic activities were calculated by plotting a standard curve using deacetylated standards provided with the kit. The unit of fluorometric absorbance used is RFU (Relative Fluorescence Unit) and was measured as;

$$\text{HDAC Activity (RFU/h/μg)} = \frac{\text{RFU(control − blank) − RFU(sample − blank)}}{\text{Reaction time(h) × protein amount added}}$$

### Small interference RNA (siRNA) transfection

THP-1 cells ($10^5$ cells/ml) were differentiated into macrophage like cells with PMA (50 ng/ml), as mentioned earlier. The PMA treated cells were then transiently transfected with 600 pmole [26] of siGENOME Human *HDAC1* (3065) siRNA–SMARTpool (Dharmacon, USA). Lipofectamine 3000 (Invitrogen, USA) was used to transfect the siRNA as per manufacturer's instructions. Briefly, the transfection reagents were mixed in RPMI media without FBS. The cells were incubated with siRNA for 24 h to allow gene silencing. Subsequently, the cells were washed and incubated with *L. donovani* at 20:1 MOI. The cells were harvested 6 h post-infection. Total RNA was extracted, followed by cDNA synthesis. The mRNA expression levels of HDAC1 and the defense genes were analyzed by qRT-PCR, as mentioned above. ON-TARGET plus Control Pool (Dharmacon, USA) was used as a negative control. The transfection efficiency was 75–80% as has also been reported by the manufacturer (Dharmacon, USA).

### Intracellular parasite load

THP1 cell line was plated at a cell density of $10^5$/ml in a six-well flat bottom plate. The cells were treated with 50 ng/ml of PMA (Sigma-Aldrich, USA) for 48 h. The adherent cells were infected with stationary-phase promastigotes at a ratio of 20:1 for 3 h. Excess non adherent promastigotes were removed by incubation of cells for 30 s in the phosphate-saline buffer (PBS). These were subsequently maintained in RPMI1640, containing 10% FBS at $37^0$ C with 5% $CO_2$ for 6 h. For visualization of intracellular parasite load, Giemsa staining was performed, and the intracellular parasite load in *L. donovani* infected THP-1 cells was then calculated [27].

### Statistical methods

GraphPad Prism (version 5.0) software (GraphPad Software, Inc.) was used for plotting data. Statistical analysis was measured using ANOVA. A $P \leq 0.05$ was considered significant [* ($P \leq 0.01$ to 0.05), ** ($P \leq 0.001$), *** ($P \leq 0.0001$), **** ($P \leq 0.0001$), ns ($P \geq 0.05$)]. Error bars used in the figures stipulate standard error of the deviation (SD).

## Results

### Defense genes expression is down-regulated in *L. donovani* infected cells

Several independent studies report differential expression of many antimicrobial peptides, like alpha, beta, and theta defensins, upon pathogen infection [26]. We, therefore, assessed the transcript levels of genes encoding antimicrobial peptides as well as genes involved in

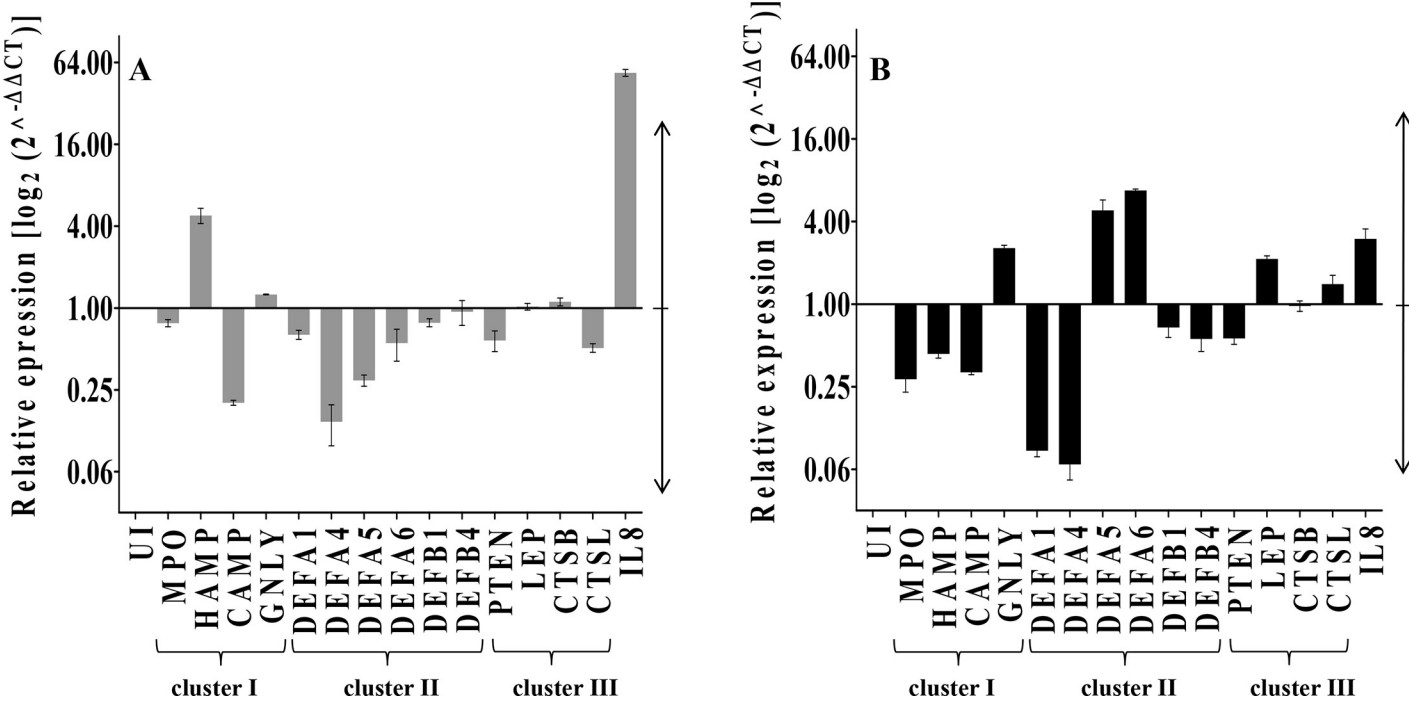

**Fig 1. Expression of host defense genes in *L. donovani* infected THP-1 cells.** THP-1 cells ($10^5$ cells/ml) were treated with PMA (50 ng/ml for 48 h), as explained in the methods section. Uninfected and infected cells were harvested at **A**: 6 h and **B**: 24 h post-infection. RNA was extracted from infected and uninfected THP-1 cells, and expression of defense genes was quantitated by qRT-PCR. Gene expression changes are depicted as fold change in transcription compared to that of uninfected cells (UI). Numbers >1 indicates up-regulation, and <1 indicates the down-regulation of host genes in infected cells as compared to the uninfected cells. *RNU6A* was used as the housekeeping gene. Data analysis was performed using a $2^{-\Delta\Delta CT}$ method. The basal level of transcript expression of uninfected cells was used for data normalization and to assess relative abundance. The results represent the mean ± SD (n = 3).

enzymatic and oxidative defense mechanisms in response to *L. donovani* infection in THP-1 cells. We focused on the transcript levels of antimicrobial factors such as *MPO*, hepcidin antimicrobial peptide (*HAMP*), cathelicidin-type antimicrobial peptide (*CAMP*), and granulysin (*GNLY*). This set of 4 genes was designated as cluster I (Fig 1A and 1B). The expression of human alpha and beta defense genes *DEFA1*, *DEFA4*, *DEFA5*, *DEFA6*, *DEFB1* and *DEFB4* were also investigated. All these defense genes are present on chromosome 8, and we have clustered them together as cluster II (Fig 1A and 1B).

We also checked the expression levels of phosphatase and tensin homolog *PTEN*, *LEP*, cathepsin B (*CTSB*), and cathepsin C (*CTSL*) as part of cluster III. *IL8*, a proinflammatory gene known to be up-regulated during *Leishmania* infection [13], was included as a positive control. THP-1 cells were infected with *L. donovani* at 1:20 multiplicity of infection (MOI), and the host cells were harvested at 6 and 24 h post-infection. The expression of the 15 defense genes was analysed by qRT-PCR and compared to the expression levels in uninfected controls. Among the 15 defense genes studied, down-regulation was observed in the majority of the defense genes at 6 h post-infection compared to that observed at 24 h post-infection.

It has been reported previously that leishmanial infection results in the down-regulation of *MPO* [12]. In the present study, also we observed the transcript levels of *MPO*, a key host defense gene [28], was down-regulated both at 6 and 24 h post-infection. Expression levels of *CAMP*, which is reported to have high leishmanicidal activity [1], showed down-regulation at both 6 and 24 h post-infection. *HAMP*, which has been reported to be up-regulated in *L. amazonensis* infected macrophages [7], showed up-regulation (~4.8-fold) at 6 h post-*L. donovani* infection. However, at 24 h post-infection, the expression of *HAMP* showed down-regulation.

The expression levels of *GNLY*, another important antimicrobial peptide [29], showed ~ 1.2-fold up-regulation at 6 h and ~ 2.5-fold up-regulation at 24 h post-infection.

Antiparasitic activities of various defensins have been reported in different leishmanial studies [11, 30, 31].We observed the down-regulation of all the defensin genes at 6 h post-infection with the exception of *DEFB4*. At 24 h post-infection, the down-regulation of *DEFB4* was observed. *DEFA5* and *DEFA6* expression showed up-regulation (~4.8-fold and ~ 6.7-fold respectively) while the other defense genes (*DEFA1*, *DEFA4*, *DEFB1* and *DEFB4*) were down-regulated at 24 h post-infection,

In cluster III, the expression levels of *PTEN*, which has been reported to get suppressed upon *L. donovani* infection [10] showed down-regulation at 6 and 24 h post-infection in the present study also. The expression of *LEP* and *CTSB* remained unchanged at 6 h post-infection, whereas *CTSL* showed down-regulation, which is in agreement with earlier reports [32]. However, at 24 h post-infection, levels of *LEP* and *CTSL* were up-regulated (~ 2.1-fold and ~ 1.4-fold respectively). Parasite infection led to the up-regulation of the expression of the pro-inflammatory gene, *IL-8* (Fig 1A and 1B) at both 6 and 24 h post-infection as has been reported earlier [26]. Taken together, we conclude that the down-regulation of antimicrobial defense genes at 6 h post-infection possibly facilitates the initial establishment of the intracellular parasite load inside the host.

## *Leishmania donovani* infection affects defense genes chromatin remodeling by the epigenetic regulator

The change in the pattern of histone-post-translational modifications is likely to have a direct effect on gene expression by affecting chromatin structure, thereby leading to highly compact chromatin conformation and limited access to transcriptional activators [16]. Epigenetic regulators mediate chromatin modification. One such histone-post-translational modification is histone acetylation and deacetylation [16]. Therefore, we hypothesized that the down-regulation of the host defense genes post-parasite infection was due to increased histone deacetylation at the promoter regions of these genes. The histone H3 acetylation (Ac-H3) patterns of defense gene promoters were investigated using chromatin immunoprecipitation (ChIP) assay. For this study, defense genes *MPO*, *HAMP*, and *GNLY* from cluster I, all the defense genes from cluster II, *PTEN* from cluster III and *IL8* were selected. Briefly, ChIP analysis was performed with chromatin of *L. donovani* infected and uninfected THP-1 cells ($10^5$ cells/ml) followed by qRT-PCR using promoter specific primers as reported in the methods section.

A decrease in Ac-H3 with *L. donovani* was observed in 8 out of 10 defense gene promoters at 6 h post-infection (Fig 2A), while at 24 h post-infection, a decrease in acetylation was observed in 7 out of 10 defense gene promoters (Fig 2B). The above data is consistent with the change in transcriptional activity at these loci during infection at 6 h and 24 h post-infection (Fig 1A and 1B). These results suggest that the silencing of key host defense genes with *L. donovani* infection could occur by epigenetic changes in the chromatin structure and post-translational modification pattern of histones.

Histone deacetylases are known to play a crucial role in regulating histone post-translational modifications, we then decided to check the occupancy of HDAC1, the enzyme that catalyses deacetylation reaction, at the promoter regions of the defense genes at 6 and 24 h post-infection using ChIP assay with HDAC1-specific antibody. At 6 h post-infection, HDAC1 occupancy at the defense genes promoters increased in 9 out of 10 genes studied (Fig 2C). These results were again consistent with the corresponding Ac-H3 occupancy data (Fig 2A). HDAC1 occupancy of *HAMP* and *IL-8* promoters were low as expected from their corresponding Ac-H3 occupancy levels (Fig 2C). At 24 h, the HDAC1 occupancy levels were lower at *GNLY*, *DEFA5*, *DEFA6*, and *IL-8* gene promoter, whereas higher for the rest of the defense

genes (Fig 2D). HDAC1 acts as a transcriptional repressor and its increased association with defense gene promoters upon *L. donovani* infection, suggests a potential role for HDAC1 in host defense genes silencing.

### Histone deacetylase expression and activity are affected in host cells post *L. donovani infection*

Since we observed an increased occupancy of HDAC1 at promoter regions of defense genes at 6 h post-infection (Fig 2C), we then decided to check the HDAC1 enzyme activity and expression in the infected THP-1 cells. THP-1 cells ($10^5$ cells/ml) infected with *L. donovani* (20:1 MOI) were harvested at 6 and 24 h post-infection. The HDAC1 protein was immunoprecipitated from the total cell lysates of these experimental samples, using an HDAC1 specific antibody.

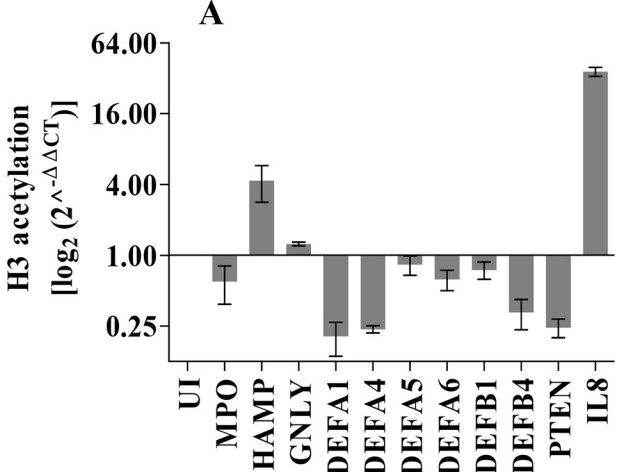
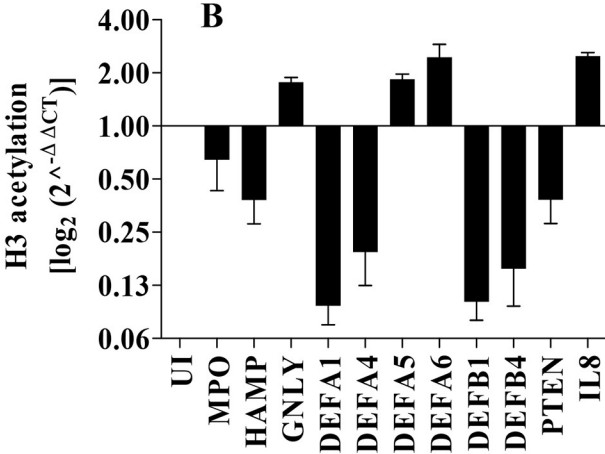
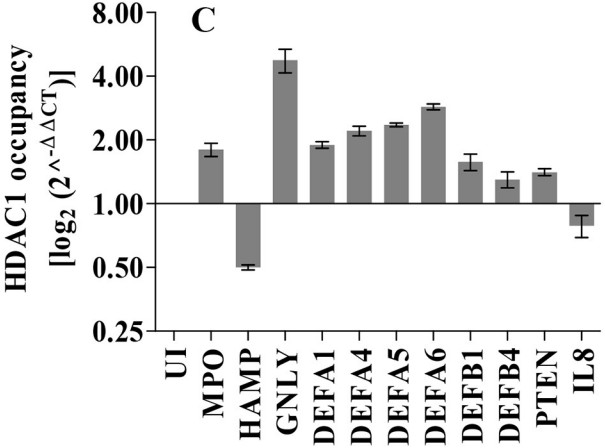
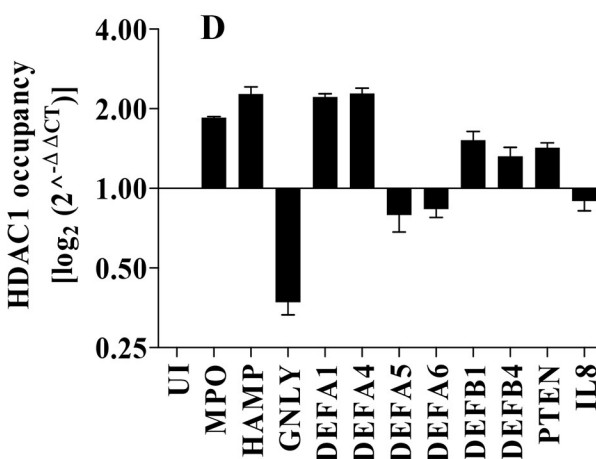

**Fig 2. Analysis of histone promoter acetylation and HDAC1 occupancy pattern of host defense genes in response to *Leishmania donovani* infection.** THP-1 cells ($10^5$ cells/ml) were infected with *L. donovani*. At 6 and 24 h post-infection, the histone acetylation pattern on the defense genes promoter was analysed by ChIP using antibodies specific for H3 acetylation **A**: 6 h post infection **B**: 24 h post-infection. HDAC1 occupancy at the defense genes promoters was also determined by ChIP using antibodies specific for HDAC1 **C**: 6 h post-infection **D**: 24 h post-infection. The immunoprecipitated DNA fragments were quantitated by qRT-PCR using primers specific for the promoter region of each of the host genes studied. The changes in the level of H3-acetylation and HDAC1 occupancy are depicted as fold change over uninfected cells (UI). Basal levels of acetylation and HDAC1 occupancy of uninfected cells were used for data normalization and assessed relative abundance. Numbers >1 indicates up-regulation, and <1 indicates down regulation of host genes in infected cells as compared to uninfected cells. The results are mean ± SD (n = 3).

The HDAC1 enzyme activity of the immunoprecipitated protein was analysed. We observed a significant increase in HDAC1 activity at 6 h post-infection when compared to the HDAC1 activity in uninfected THP1 cells ($P = 0.00001$). HDAC1 activity at 24 h post-infection was down to basal levels as in the uninfected THP1 cells (Fig 3A).

We next examined whether the expression levels of HDAC1 is altered in response to parasite infection. Gene expression of *HDAC1* in THP-1 cells infected with *L. donovani* was evaluated using HDAC1 specific primers, as reported in the methods section. The transcript levels

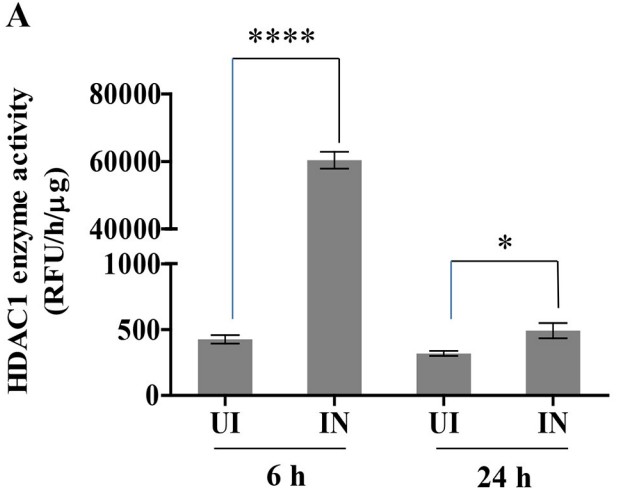

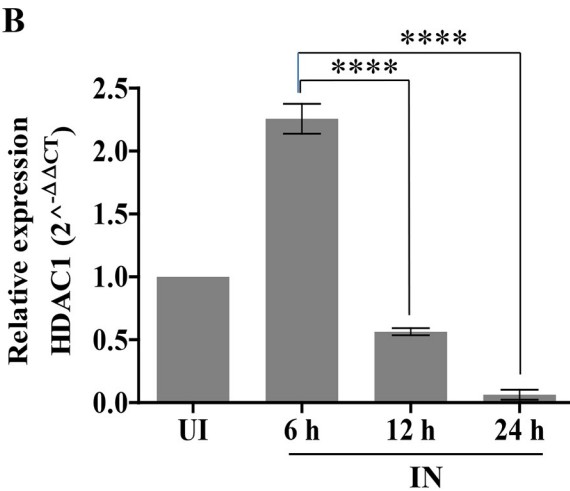

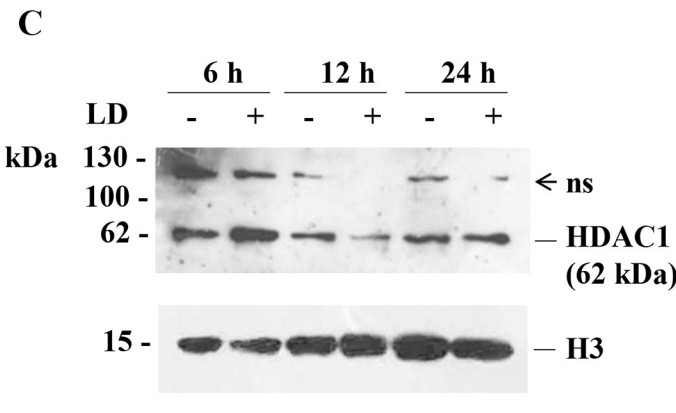

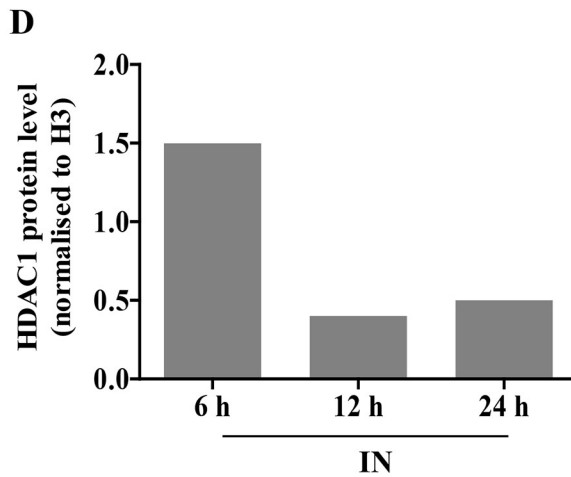

**Fig 3. Histone deacetylase 1 (HDAC1) expression and enzyme activity in host cells post *L. donovani* infection.** THP-1 cells ($10^5$cells/ml) were infected with *L. donovani* (20:1, MOI). The cells were harvested at different time points. **A:** 6 and 24 h post-infection HDAC1 was immunoprecipitated from infected and unifected host cells, and lysates were analysed for HDAC1 enzyme activity by fluorescent assay kit. The unit of fluorometric absorbance used is RFU (Relative Fluorescence Unit) and was calculated, as mentioned in the methods section. **B:** RNA was extracted from uninfected (UI) and infected cells (IN) at 6 h, 12 h and 24 h post-infection. *HDAC1* expression was quantitated by qRT-PCR. Fold change values of *HDAC1* mRNA is represented relative to uninfected condition (UI). Basal levels of *HDAC1* expression in uninfected cells was used for data normalization and was taken as 1.0. **C:** Nuclear extracts of uninfected and infected THP-1 cells were harvested at 6, 12 and 24 h post-infection and lysed. Western blotting was performed using HDAC1 specific antibodies to observe HDAC1 protein expression. The symbols (-) and (+) represent uninfected and infected conditions respectively. H3 was used as housekeeping control to confirm equal loading of protein. Molecular weight markers are mentioned at left hand side of the blot and 'ns' refers to non-specific protein bands—**D:** Densitometric analysis of western blot using ImageJ software and normalized to H3 levels. The results represent the mean ± SD (n = 3). ANOVA was used to determine statistical significance. P-value for significance: * ($P{\leq}0.01$ to $0.05$), **** ($P{\leq}0.00001$).

of *HDAC1* at 6 h post-infection was ~2.2-fold ($P<0.0001$) higher compared to the levels in the uninfected cells (UI). A decrease in *HDAC1* expression was observed at 12 and 24 h post-infection in comparison to the values obtained at 6 h post-infection (Fig 3B). Taken together our results indicate that increased expression of HDAC1 in macrophages infected with *L. donovani* is in part due to the transcriptional control.

HDAC1 has been reported as a transcriptional regulator for several potential tumor suppressor genes and antimicrobial peptides [17, 26]. Western blot (Fig 3C), followed by densitometric analysis (Fig 3D), was done to study the expression of HDAC1 in the nuclear fractions of infected and uninfected cells. At 6 h post-infection, higher expression of HDAC1 protein (~ 1.5 fold) was observed compared to the levels observed in uninfected cells (UI). A decrease in HDAC1 levels was observed at 12 and 24 h post-infection in comparison to the values obtained at 6 h post-infection (Fig 3C). Our data indicate that *Leishmania* infection results in increased expression of HDAC1 at 6 h post-infection both at the transcriptional and translational levels.

### *HDAC1* silencing up-regulates defense gene expression and impairs the ability of *L. donovani* to propagate intracellularly in THP-1 host cells

The expression of defense genes and intracellular parasite load were analysed after silencing the *HDAC1* expression with siRNA to determine if the expression of host *HDAC1* is essential. THP-1 cells were transfected with *HDAC1*-siRNA or scrambled-siRNA (Sc-siRNA) at 24 h before infection with *L. donovani*. After 6 h post-infection, the infected and uninfected cells were harvested, followed by RNA isolation, cDNA synthesis, and qRT-PCR, as mentioned in the methods section. The transcript level of *HDAC1* in Sc-siRNA transfected cells was higher (~ 1.5-fold) in infected cells in comparison to the uninfected cells (Fig 4A). These results show that scrambled siRNA has no inhibitory effect on host *HDAC1* expression. In the case of *HDAC1*-siRNA transfected cells, a decrease in *HDAC1* expression was observed in infected cells when compared to the uninfected cells. Further, the expression of *HDAC1* in infected cells transfected with *HDAC1*-siRNA showed significant inhibition ($P = 0.00005$) in comparison to cells transfected with scrambled-siRNA (Fig 4A). This data confirms a specific silencing effect of *HDAC1*-siRNA on the expression of host *HDAC1*.

We next studied the parasite load within the infected host cells transfected with Sc-siRNA and *HDAC1*-siRNA (Fig 4B). Parasite load was analysed by visually counting the amastigotes within the infected THP1 cells transfected with siRNA, after Giemsa staining. We observed a significant decrease ($P<0.00001$) in the parasite load in cells transfected with *HDAC1*-siRNA as compared to the cells transfected with Sc-siRNA. The result suggests an important role of HDAC1 in the intracellular survival of the parasite.

We further checked if *HDAC1* has a direct role in silencing host defense genes. To this end, the expressions of five defense genes (*DEFA1*, *DEFA5*, *DEFA6*, *DEFB4*, and *MPO*) were analyzed in the infected cells transfected with *HDAC1*-siRNA. Expression of *DEFA1* (~20-fold, $P = 0.0012$), *DEFA5* (~55-fold, $P = 0.0002$), *DEFA6* (~2.8-fold, $P = 0.0015$), *DEFB4* (~21-fold, $P = 0.0029$) and *MPO* (~31-fold, $P<0.0001$) were up-regulated in infected THP-1 cells transfected with HDAC1-siRNA as compared to the levels in cells transfected with scrambled-siRNA (Fig 4C). Collectively, these results indicate the role of HDAC1 in host defense gene-silencing and thereby enabling the intracellular survival of the parasite.

### HDAC1 inhibition by pharmacological inhibitors impairs the ability of *L. donovani* to propagate intracellularly

Sodium butyrate (NaB), a short chain fatty acid that blocks class I and II HDACs (36), was examined for its role in HDAC1 inhibition and *L. donovani* infection. A dose dependent

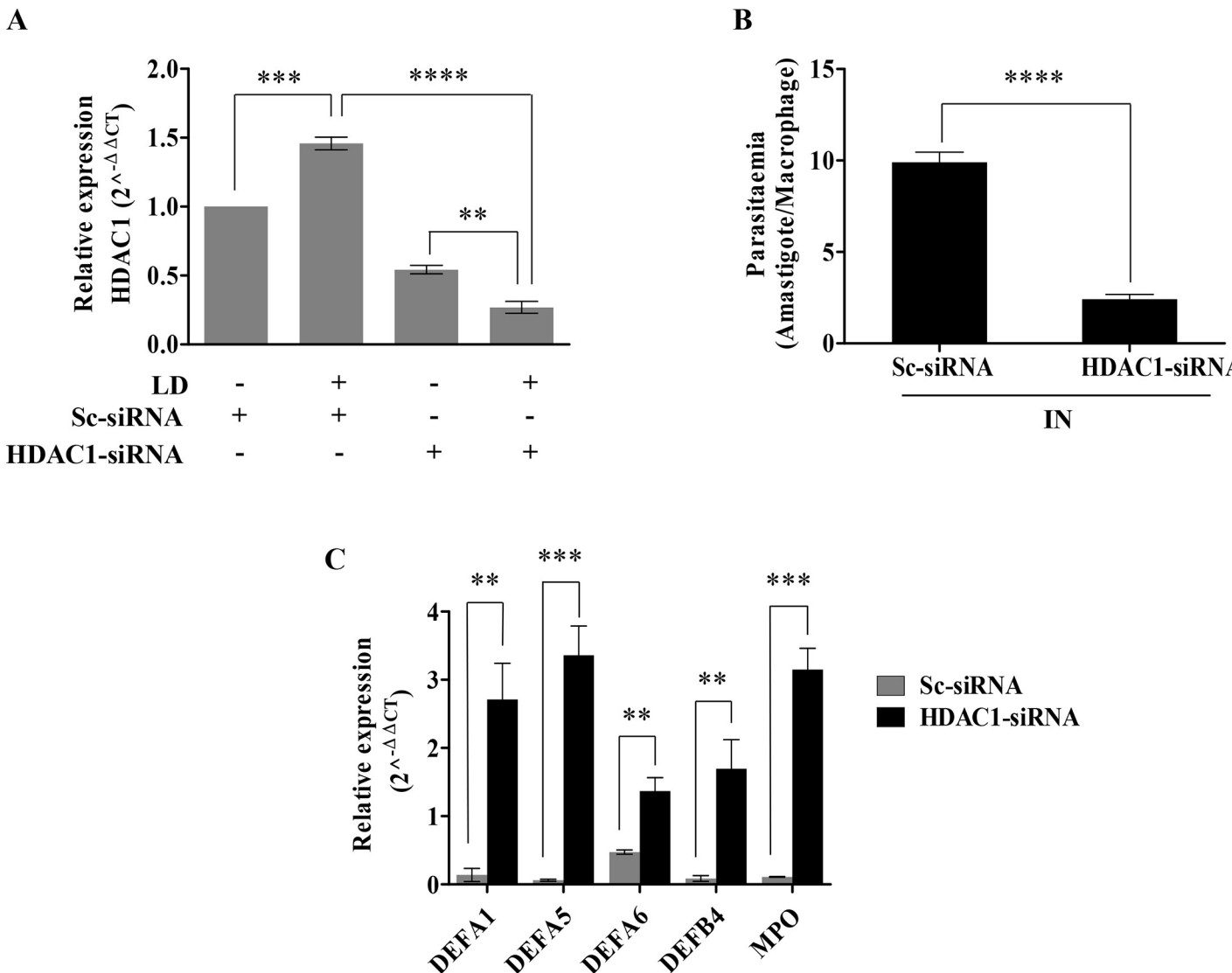

**Fig 4. Effect of *HDAC1* silencing on the expression of host defense genes and intracellular parasite load in *L. donovani* infected THP-1 cells.** THP-1 cells (10$^5$ cells/ml) were transfected with *HDAC1*-siRNA (600 pmol) using lipofectamine. Scrambled-siRNA transfected THP1 cells were used as a control. After 24 h of transfection, the THP-1 cells were infected with *L. Donovani* (20:1 MOI). The cells were then harvested at 6 h post-infection, and RNA was extracted, followed by qRT-PCR to determine the impact of *HDAC1* silencing on host defense genes and intracellular parasite load. **A:** Expressions of *HDAC1*were quantified by qRT-PCR. Fold change values of *HDAC1* mRNA is represented relative to uninfected-Sc-siRNA transfected condition. The basal level of *HDAC1* in uninfected-Sc-siRNA transfected cells was used for data normalization and was taken as 1.0. **B:** THP1 cells were transfected with *HDAC1*-siRNA or Sc-siRNA for 24 h and were then incubated with *L. donovani* (IN) for 3 h. After 6 h post-infection, cells were stained with Giemsa and the number of infected cells and the number of amastigotes were counted visually. **C:** Expression of defense genes in infected THP-1 cells transfected with *HDAC1*-siRNA or Sc-siRNA was determined by qRT-PCR.Results represent mean ± SD (n = 3). Significance was calculated by ANOVA, ** (*P*≤.01), *** (*P*≤0.001) and **** (*P*≤0.00001).

inhibition of HDACs by NaB and its impact on parasite survival was next evaluated. THP-1 cells were first infected with *L. donovani* followed by treatment with the increasing concentration of NaB for 6 h. The transcript levels of *HDAC1* were quantitated by qRT-PCR at 6 h post-infection and NaB treatment. The expression of *HDAC1* decreased with an increasing concentration of NaB (1.25–20 mM) (Fig 5A). Total HDAC enzyme activity was determined using nuclear extracts from infected THP-1 cells treated with an increasing concentration of NaB (1.25–10 mM) (Fig 5B). A decrease in transcript levels of *HDAC1* (*P*<0.001) and total HDAC

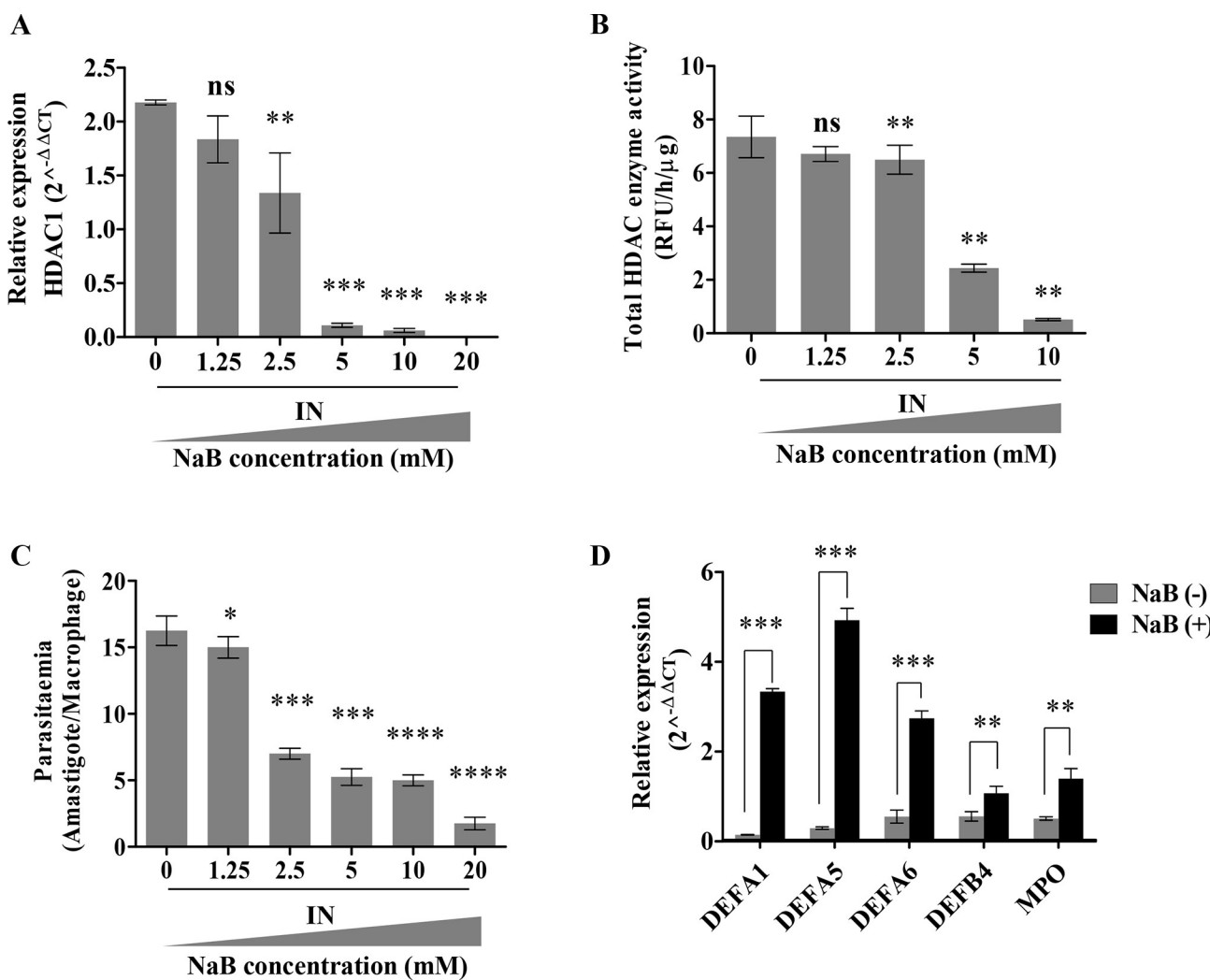

**Fig 5. Effect of inhibition of HDACI activity by pharmacological inhibitor NaB on defense gene expression and parasite load in *L. donovani* infected THP-1 cells.** THP-1 cells ($10^5$cells/ml) were incubated with *L. donovani* (20:1 MOI) for 3 h. Subsequently, the infected (IN) cells were incubated with increasing amounts of sodium butyrate (NaB) and were harvested at 6 h. **A:** Expression of *HDAC1* was measured by qRT-PCR and transcription levels were plotted. The significance of the difference in expression levels was measured in infected (IN) and untreated samples (NaB concentration– 0 mM). **B:** Total HDAC activity in nuclear extracts of infected cells in the presence of increasing amounts of NaB was detected by fluorescent assay kit. The unit of fluorometric absorbance used is RFU (Relative Fluorescence Unit) and was calculated as mentioned in the methods section. The significance of the difference in activity was measured in infected (IN) and untreated samples (NaB concentration– 0 mM). **C:** Dose dependent impact of HDAC1 inhibitor on the parasitaemia count. Infected THP-1 cells (IN) were incubated in increasing concentrations of NaB. After 6 h, cells were stained and amastigotes enumerated visually. **D:** Expression levels of the host defense genes, post-infection in the presence and absence of NaB (5 mM) were quantified by qRT-PCR. Results represent the mean ± SD (n = 3). The statistical significance was determined using ANOVA, ns ($P$>0.05),* ($P$≤0.01 to 0.05), ** ($P$≤0.01), *** ($P$≤0.001), *** ($P$≤0.0001) and **** ($P$≤0.00001).

activity ($P$<0.001) was observed with 5 mM of NaB in comparison to untreated infected cells. Infected cells treated with NaB were further analysed for parasite load by visually counting the amastigotes within the THP1 cells after Giemsa staining. A decrease in parasite load within infected macrophages was observed with increasing concentrations of NaB (Fig 5C). To check the toxicity of NaB on the parasite alone, we incubated the promastigotes ($10^8$ cells) with 10 mM of the inhibitor. NaB treatment had no inhibitory effect on the promastigotes (S1A Fig).

We further evaluated if inhibition of HDAC1 plays a critical role in the expression of host defense genes. We observed an up-regulation in the transcript expression of four host defense

genes *DEFA1* (~21-fold, *P*<0.001), *DEFA5* (~16-fold, *P*<0.001), *DEFA6* (~4-fold, *P*<0.001), *DEFB4* (~1-fold, *P* = 0.0087) and *MPO* (~1.8-fold, *P* = 0.0025) in infected macrophages in the presence of 5 mM of NaB in comparison to untreated THP-1 cells (Fig 5D).

Vorinostat, also termed as suberoylanilidehydroxamic acid (SAHA) is a known class I and class II HDAC inhibitor [33]. It is FDA approved [34] for the treatment of cutaneous T cell lymphoma (CTCL) and Sezary syndrome [35, 36]. It is also being evaluated for the treatment of various other oncology and HIV infection [37].

A dose dependent inhibition of HDACs by SAHA and its impact on parasite survival was next evaluated. THP-1 cells were first infected with *L. donovani* followed by treatment with the increasing concentration of SAHA (0–10 μM) for 6 h. The cells were then harvested and the transcript levels of host *HDAC1* were quantitated by qRT-PCR. As in the case of NaB, the presence of SAHA also resulted in a concentration dependent (0–10 μM) decrease in *HDAC1* transcript levels (*P* = 0.0005), with 5 μM SAHA (Fig 6A).

We further validated the impact of SAHA on parasite survival within the host by visually counting the amastigotes after Giemsa staining. A concentration dependent inhibition of the parasite load within the host cells treated with SAHA was observed (Fig 6B). We observed a decrease in the parasite load (*P*<0.0001) with 5 μM SAHA (Fig 6B). The promastigotes ($10^8$ cells) were incubated with 5 μM of SAHA to check the effect of SAHA on the parasite. No inhibitory effect of 5 μM SAHA was observed on the promastigotes (S1B Fig). To understand if the mechanism of action of SAHA and NaB is similar, we checked the transcript levels of *DEFA1*, *DEFA5*, *DEFA6*, *DEFB4*, and *MPO* defense genes in infected THP-1 cells in the presence of 5 μM SAHA. As expected, an increase in the expression levels of *DEFA1* (~21-fold, *P*<0.0001), *DEFA5* (~6-fold, *P*<0.0001), *DEFA6* (~6-fold, *P*<0.0001), *DEFB4* (~16-fold, *P* = 0.0079) and *MPO* (~1.5-fold, *P*<0.0001) were observed in the infected THP-1 cells in the presence of 5 μM SAHA as compared to the levels in the absence of SAHA (Fig 6C).

Our data confirm the role of HDAC1 in host-defense gene silencing and host cell enabling the parasite survival. Since both SAHA and NaB inhibit not only HDAC1 but also other class I HDACs, our data on siRNA resulting in specific silencing of HDAC1 and leading to intracellular parasite survival further confirms the role of HDAC1 in host defense.

## Discussion

*Leishmania donova*ni, an intracellular protozoan parasite, results in an array of diseases in the human hosts. The parasite has evolved sophisticated mechanisms to manipulate the host response. The hallmark of survival of the *Leishmania* parasite in macrophages is the inhibition of the activation of the host cell's innate immune response, thereby enabling the parasite to survive and replicate within the macrophages. Multiple mechanisms may be involved in the inhibition of macrophage function by intracellular infections. Like other prokaryotic and viral pathogens, an intracellular protozoan parasite, *Leishmania*, also leads to manipulation of the host cells via epigenetic modifications of the host genome [10, 38]. Epigenetics involves three main processes: DNA methylation [3], histone modification [3, 4] and non-coding RNA-associated gene silencing [39].

Genome-wide methylation of the host-macrophage genome after *Leishmania* infection showed epigenetic modifications of the host-cell and significant changes in the CpG sites of several genes involved in host-defense by macrophage activation [38]. Interestingly, decreased methylation of HDAC4 correlated with increased mRNA expression [38]. Earlier reports indicated increased levels of HDAC1 in *L. amazonensis* infected macrophages [40]. Parasites utilize multiple pathways and mechanisms to mediate epigenetic changes and in turn, control the host defense response to infection.

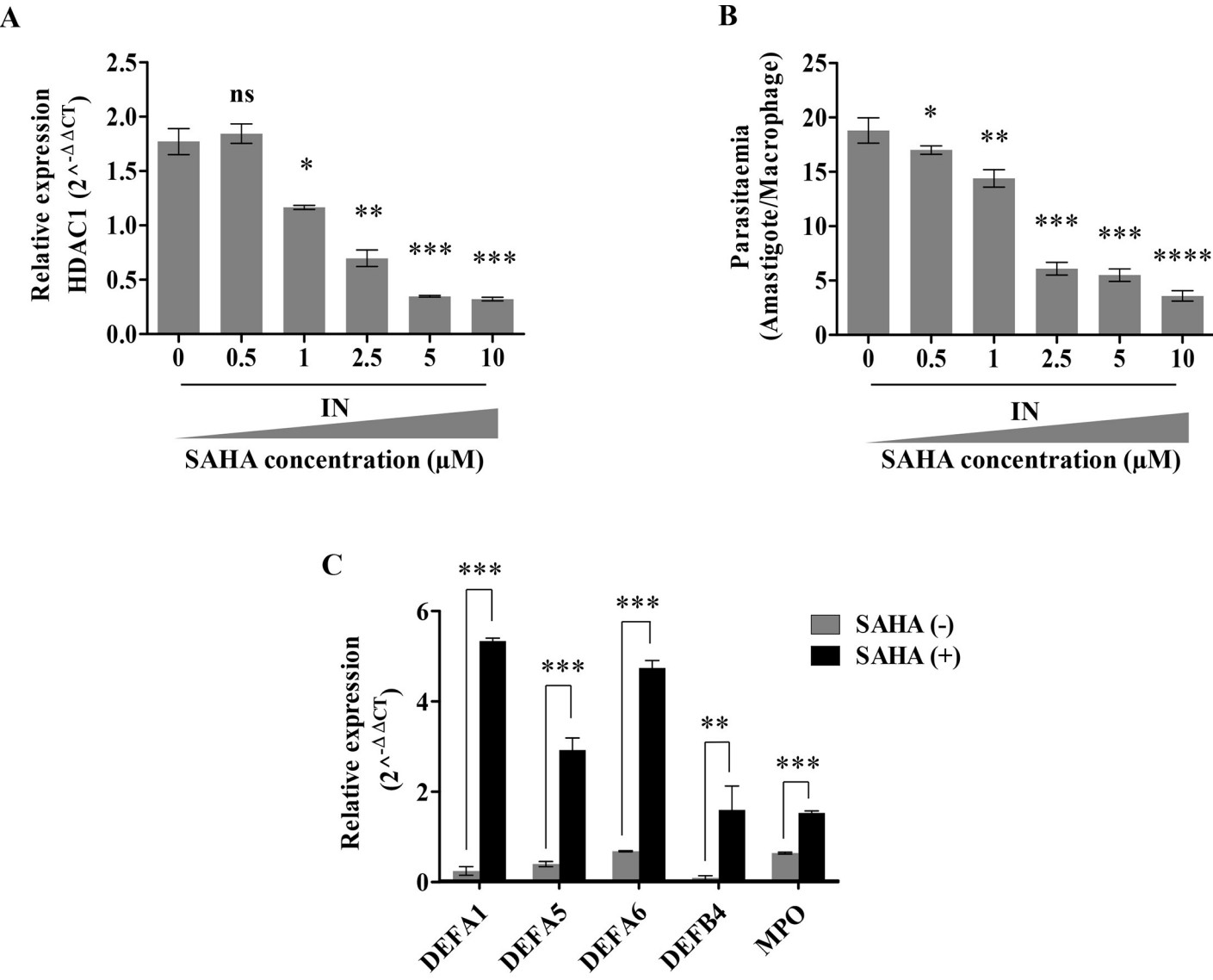

**Fig 6. HDAC1 inhibiting drug SAHA impairs parasite survival.** THP-1 cells ($10^5$ cells/ml) were infected with *L. donovani* (20:1 MOI) for 3 h.The infected (IN) cells were then incubated with increasing amounts of SAHA and were then harvested at 6 h. **A:** Expression of *HDAC1* infected (IN) THP-1 cells after treatment with different concentrations of SAHA. Significance was measured concerning infected and untreated samples (SAHA concentration– 0 uM). **B:** Dose-dependent impact of HDAC1 inhibitor, SAHA, on the parasitemia count. Infected THP1 cells (IN) were incubated with increasing concentrations of SAHA for 6 h. Cells were stained after 6 h and amastigotes enumerated visually. **C:** The expression levels of the host defense genes, 6 h post-infection in the absence and presence of SAHA (5 μM) were quantified by qRT-PCR. Results represent the mean ± SD (n = 3). The statistical significance was determined using ANOVA, ns (*P*>0.05), * (*P*≤0.05), ** (*P*≤0.01), *** (*P*≤0.001), *** (*P*≤0.0001) and **** (*P*≤0.00001).

The work presented here demonstrates a mechanism by which *L. donovani* controls host cell gene expression and function based on a modification of the epigenome. In our work, we have observed the down-regulation of defense genes upon *L. donovani* infection, which is in agreement with previous reports [6, 29, 32]. The underlying molecular mechanisms that assist such genome modifications remain mostly unknown.

We investigated the role of HDAC1 in chromatin modulation and its impact on host defense genes in response to *L. donovani* infection. HDAC1 expression and activity were found to be dependent upon the stage of infection. At an earlier time, 6 h post-infection, the

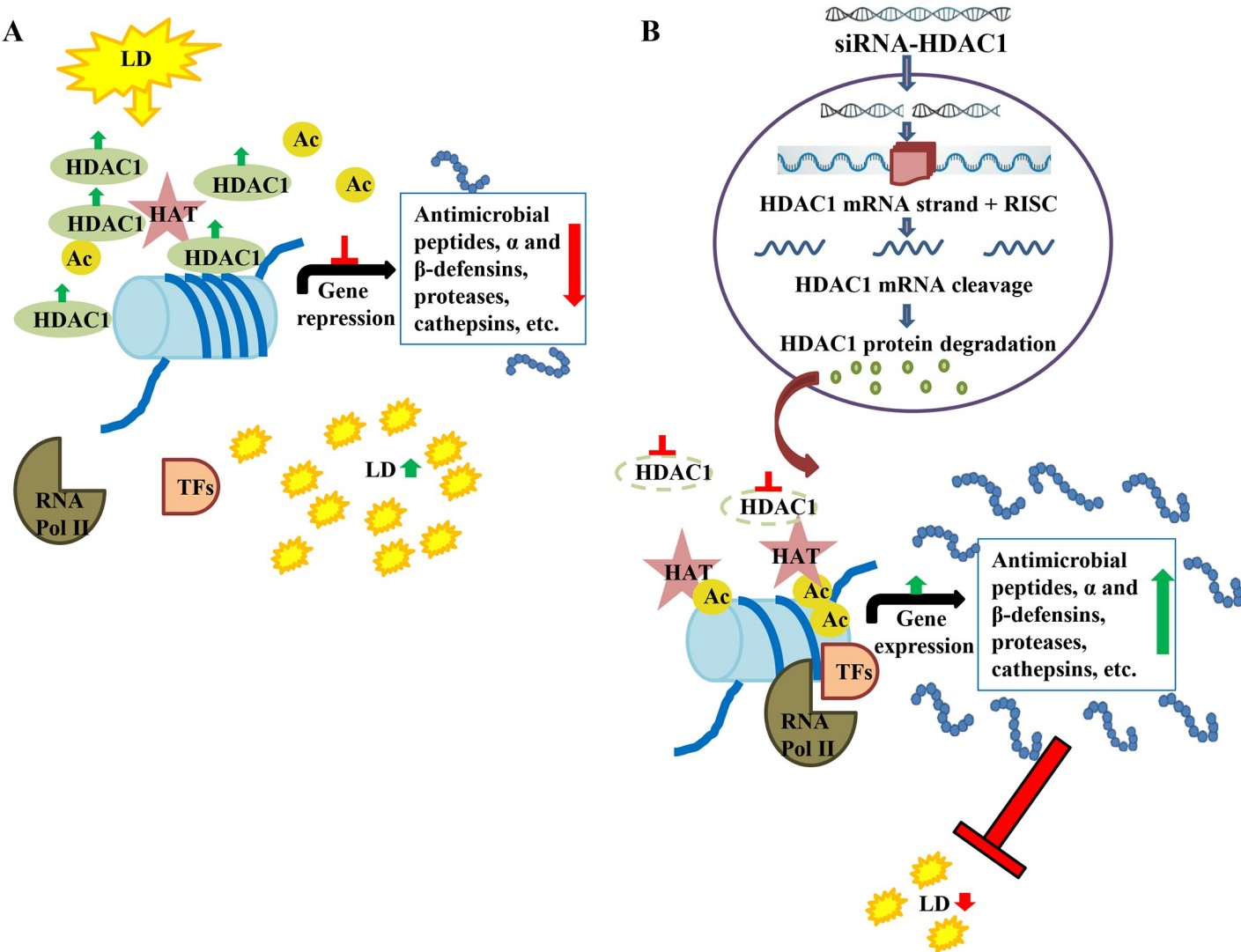

**Fig 7. Mechanism of action of siRNA mediated HDAC1 down-regulation and its impact on intracellular *L. donovani* A:** Infection with *L. donovani* (LD), increases histone deacetylase (HDAC) levels which further leads to decrease in expression of the antimicrobial factors. Thereby facilitating the establishment of parasite infection within the host cells. **B:** Transfection of host cells with HDAC1-siRNA, leads to reduced HDAC1 and reduced deacetylation of histones, facilitating increased transcription of host defense genes and decreased intracellular load of *L. donovani*.

expression of HDAC1 was significantly higher, suggesting the importance of the down-regulation of host innate defense genes during the preliminary establishment of the parasite infection. The expression levels of *HDAC1* were further investigated to find a correlation with the acetylation of promoter regions of various host defensins. The defense genes that had a low expression on infection also had low levels of acetylated histones bound to their promoter regions, as observed by ChIP assay. HDAC1 acts as a transcriptional repressor and its increased association with defense gene promoters upon *L. donovani* infection, suggests a potential role for HDAC1 in host defense genes silencing. Earlier reports also indicate that HDACs help in host gene silencing in response to parasite infection [7, 10, 12].

In the present study, Pharmacological inhibitors and siRNA resulted in inhibition of HDAC1, significantly leading to increased expression of defense genes and decreased the levels of parasite load within the infected host cells. These results will facilitate the identification of

novel drug targets and at the same time, would add to the existing knowledge and lead to further possibilities to suppress the parasite growth.

We have summarized the epigenetic modulation of host genes by *L. donovani* in a diagrammatic model (Fig 7A and 7B). Histone acetylases (HATs) are reported to result in uncoiling of chromatin through acetylation (Ac) of histone, thereby exposing promoters of antimicrobial factors, defensins, proteases, cathepsins, etc., thus, in turn, aiding in their transcription [16]. Infection with *L. donovani* (LD) increases histone deacetylase (HDAC) levels leading to decreased acetylation. Decreased acetylation resulted in down-regulation in the expression of antimicrobial peptides leading to the establishment of parasite infection in the host cells (Fig 7A). Transfection of host cells with siRNA against HDAC1 resulted in the down-regulation of expression of HDAC1-mRNA and HDAC1 protein levels (Fig 7B), possibly through RNA-induced silencing complex (RISC) mediated mRNA degradation [41]. Inhibition of HDAC1 levels leads to an increase in gene expression of antimicrobial factors and defensin genes in the host cells leading to a decreased intracellular load of *L. donovani*.

## Supporting information

**S1 Table. Genes and primers used in this study.**
(DOCX)

**S1 Fig. NaB and SAHA has no inhibitory effect on promastigotes. A.** Promastigotes were pre-treated with 0 or 10 mM of NaB prior to incubation with THP-1 cells for parasite infection (MOI– 20:1). **B.** Promastigotes were pre-treated with 0 or 5 μM of SAHA prior to incubation with THP-1 cells for parasite infection (MOI– 20:1).
(DOCX)

## Acknowledgments

We thank the Central Instrumentation Facility at the School of Life Sciences, Jawaharlal Nehru University, for providing instrumentation facility. RMB is A S Paintal Distinguished Scientist Chair of ICMR.

## Author Contributions

**Conceptualization:** Rohini Muthuswami, Rentala Madhubala.

**Funding acquisition:** Rohini Muthuswami, Rentala Madhubala.

**Investigation:** Gargi Roy, Harsimran Kaur Brar.

**Methodology:** Gargi Roy, Rentala Madhubala.

**Supervision:** Rohini Muthuswami, Rentala Madhubala.

**Writing – original draft:** Gargi Roy, Rohini Muthuswami, Rentala Madhubala.

**Writing – review & editing:** Rohini Muthuswami, Rentala Madhubala.

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
