## [Decision Letter · Decision Letter 0]

14 Nov 2019

Dear Dr. Muthuswami:Thank you very much for submitting your manuscript "Epigenetic Regulation of Defense Genes by Histone Deacetylase 1 in Human Cell Line-Derived Macrophages Promotes Intracellular Survival of Leishmania donovani." (#PNTD-D-19-01539) for review by PLOS Neglected Tropical Diseases. Your manuscript was fully evaluated at the editorial level and by independent peer reviewers. The reviewers appreciated the attention to an important problem, but raised some substantial concerns about the manuscript as it currently stands. These issues must be addressed before we would be willing to consider a revised version of your study. We cannot, of course, promise publication at that time.We therefore ask you to modify the manuscript according to the review recommendations before we can consider your manuscript for acceptance. Your revisions should address the specific points made by each reviewer. When you are ready to resubmit, please be prepared to upload the following:(1) A letter containing a detailed list of your responses to the review comments and a description of the changes you have made in the manuscript.(2) Two versions of the manuscript: one with either highlights or tracked changes denoting where the text has been changed (uploaded as a "Revised Article with Changes Highlighted" file); the other a clean version (uploaded as the article file).(3) If available, a striking still image (a new image if one is available or an existing one from within your manuscript). If your manuscript is accepted for publication, this image may be featured on our website. Images should ideally be high resolution, eye-catching, single panel images; where one is available, please use 'add file' at the time of resubmission and select 'striking image' as the file type. Please provide a short caption, including credits, uploaded as a separate "Other" file. If your image is from someone other than yourself, please ensure that the artist has read and agreed to the terms and conditions of the Creative Commons Attribution License at http://journals.plos.org/plosntds/s/content-license (NOTE: we cannot publish copyrighted images). (4) If applicable, we encourage you to add a list of accession numbers/ID numbers for genes and proteins mentioned in the text (these should be listed as a paragraph at the end of the manuscript). You can supply accession numbers for any database, so long as the database is publicly accessible and stable. Examples include LocusLink and SwissProt.(5) To enhance the reproducibility of your results, we recommend that you deposit your laboratory protocols in protocols.io, where a protocol can be assigned its own identifier (DOI) such that it can be cited independently in the future. For instructions see http://journals.plos.org/plosntds/s/submission-guidelines#loc-methodsPlease note while forming your response, if your article is accepted, you may have the opportunity to make the peer review history publicly available. The record will include editor decision letters (with reviews) and your responses to reviewer comments. If eligible, we will contact you to opt in or out.While revising your submission, please upload your figure files to the Preflight Analysis and Conversion Engine (PACE) digital diagnostic tool, https://pacev2.apexcovantage.com/ PACE helps ensure that figures meet PLOS requirements. To use PACE, you must first register as a user. Then, login and navigate to the UPLOAD tab, where you will find detailed instructions on how to use the tool. If you encounter any issues or have any questions when using PACE, please email us at figures@plos.org.We hope to receive your revised manuscript by Jan 13 2020 11:59PM. If you anticipate any delay in its return, we ask that you let us know the expected resubmission date by replying to this email.To submit a revision, go to https://www.editorialmanager.com/pntd/ and log in as an Author. You will see a menu item call Submission Needing Revision. You will find your submission record there. Sincerely,Armando Jardim, PhDAssociate EditorPLOS Neglected Tropical DiseasesHans-Peter FuehrerDeputy EditorPLOS Neglected Tropical Diseases***********************Reviewer's Responses to Questions**Key Review Criteria Required for Acceptance?**

**Methods**

-Are the objectives of the study clearly articulated with a clear testable hypothesis stated?

-Is the study design appropriate to address the stated objectives?

-Is the population clearly described and appropriate for the hypothesis being tested?

-Is the sample size sufficient to ensure adequate power to address the hypothesis being tested?

-Were correct statistical analysis used to support conclusions?

-Are there concerns about ethical or regulatory requirements being met?

Reviewer #1: The authors have addressed the role of the chromatin remodeling enzyme HDAC1 in the downregulation of defensin genes. Overall, the experiments are well designed, although there are some important concerns on the methodology applied to the study and the statistics applied by the authors.Reviewer #2: it is very well designed, only I have a comment; why you didn't use ATCC organism.Is the strain you used typed already or no?Reviewer #3: When taken at face-value the results presented in this manuscript support the hypothesis that Leishmania infection of macrophages triggers histone deacetylase 1 (HDAC1)-mediated down-regulation of defense genes, thereby promoting intracellular parasite survival and growth. In general, the methodology used (mostly qRT-PCR analysis of mRNA and ChiP samples) are suitable to address the questions asked.--------------------**Results**

-Does the analysis presented match the analysis plan?

-Are the results clearly and completely presented?

-Are the figures (Tables, Images) of sufficient quality for clarity?

Reviewer #1: The overall quality of the figures is not good. There is an evident lack of resolution in the graphics.The authors should improve the figures. Importantly, the statistical interpretation based on T student test seems not to be correct. ANOVA is highly recommended.Reviewer #2: I recumbent the author to add a picture related to the mechanism of action of siRNAReviewer #3: However, numerous problems with data presentation and the use of imprecise language throughout significantly undermine confidence in the validity of the results obtained. In particular, several figures are mis-labeled and there are numerous questions as to whether appropriate controls are used. An over-reliance in using the ratio of infected to uninfected samples to present most data is of particular concern (see below). In addition, several examples of inconsistent results between experiments are ignored.--------------------**Conclusions**

-Are the conclusions supported by the data presented?

-Are the limitations of analysis clearly described?

-Do the authors discuss how these data can be helpful to advance our understanding of the topic under study?

-Is public health relevance addressed?

Reviewer #1: The authors discuss very poorly their results. They should discuss the observed modulation of defensin gene expression resulted from the experimental time course and the possible impact in the infection.Reviewer #2: very well writtenReviewer #3: There is very little discussion of alternative explanations for the observed results. However, the authors do a good job discussing the possible relevance of the findings to public health.--------------------**Editorial and Data Presentation Modifications?**

Use this section for editorial suggestions as well as relatively minor modifications of existing data that would enhance clarity. If the only modifications needed are minor and/or editorial, you may wish to recommend “Minor Revision” or “Accept”. Reviewer #1: The authors based the work on previously published studies, that demonstrated the correlation of HDAC1 induced expression by pathogens and the repression of some innate responsive genes.The figures have poor quality and the statistical analysis seems inappropriate. Limes 63 and 64, the authors make a general assumption regarding the gene silencing induced by pathogens. This assumption is too general and it is not applied in many models.Regarding the methodology, the authors use a general acetyl lysine antibody in the experiments. They should have used a more precise mark for Histone acetylation in the assays. By no means, Figure 3 presents specific HDAC1 activity data. Did the have immunoprecipitated HDAC1 prior to the activity measurement?There is no reason for the use of NAB in the experiments. NaB is a pan HDAC inhibitor and the results are not really useful. Instead, they should have used another HADC1 specific inhibitor. The authors should have shown the level of transfection, the percentage of transfected cells when they proceeded with the si experiments. Figure 4 shows an important effect of scrambled si on the parasite load. Please, explain.Reviewer #2: To elaborate more about the strain that has been used from Dr Dr StephenBeverlyTo add a picture related to the mechanism of action of siRNAReviewer #3: Much of the results section is spent merely re-stating the results from the figures, rather than summarizing what they (might) mean. In addition, the much of the language used lacks precision. I am particularly concerned about the relative modest fold-change value (often less than 2-fold) and the lack of negative controls (i.e. genes that are not expected to change expression). How can were be sure the results shown don't merely represent noise. In this context, it is important to know whether the three replicates used in most experiments represent three independent samples or merely technical replicates from the same sample? The following comments are meant to illustrate some (but by no means all) of the problems with the Results section.Line 280: there appears to be text missing after "upon pathogen"Lines 285-291: what is the logic behind the using the terms Cluster I, Cluster II and Cluster III to describe the genes studied? They certainly don't cluster by their behavior.Figure 1: How do fold-change values of <1 indicate up-regulation and >1 indicate down-regulation (lines 297-298)? The y-axis indicates that the fold-change is presented on a log2-scale, but this is clearly not the case. the results would be much clearly if that were the case. On the x-axis, one gene is labeled LKP, while in the text it is called LEP. Why is there no discussion of the (almost complete) lack of correlation between the fold-changes at 6 hr and 24 hr. Line 307: DEFB4 is not down-regulated Line 319-321: there is no experimental evidence that down-regulation of the defense gene mRNA levels allows sustained inhibition of antimicrobial defense and facilitates intracellular survival of the parasite. This is merely the hypothesis.Line 324-326: It would be better to provide some context/background for this statement and hypothesis.Figure 2: The same scale (preferably log2) should be used for all plotsFigure 3: While it is obvious that + and - in panel B present uninfected and infected cells, respectively, this is not explicitly stated in the legend. I find panel D (and its description in the text) to be very confusing. It is not clear how you can have a negative value for HDAC enzyme activity. In my option, it would be much better to present the HDAC levels relative to the 0 hr sample in panel D (as it is in panel C). It is also not clear what sample values were compared in the statistical analyses. Figure 4: In general, I find use of the infected/uninfected ratio confusing in presenting these data. In my option, it is more important to directly compare the results obtained in with and without siRNA. This is particularly true for panel B, where the value in the uninfected sample is very low (since they contain no kDNA) and could introduce large errors. It would be better to present these data as ratio to those obtained with a know number of parasites. I am also concerned about the use of the "scrambled-siRNA" as the control since it is clearly different from the no siRNA (negative) control. Figure 5 and Figure 6: Once again, use of the infected/uninfected ratio is confusing, since the inhibitors are almost certainly doing something in the uninfected cells. In my option, it would be much more informative to present all the date relative to untreated uninfected cells. The data presented in panel D of both figures is not entirely consistent with the values presented in Fig 1. Indeed, the results in panel 5A and 6A show considerable variation in the (relative) HDAC1 levels in the absence of inhibitor. Line 510: "hallmark" is one word Line 527: I hesitate to call the down-regulation of defense gene mRNA levels as "major", since it was only 3-fold at most (I.e. DKFA4 in Fig. 1A).--------------------**Summary and General Comments**

Use this section to provide overall comments, discuss strengths/weaknesses of the study, novelty, significance, general execution and scholarship. You may also include additional comments for the author, including concerns about dual publication, research ethics, or publication ethics. If requesting major revision, please articulate the new experiments that are needed.Reviewer #1: The authors based the work on previously published studies, that demonstrated the correlation of HDAC1 induced expression by pathogens and the repression of some innate responsive genes.The figures have poor quality and the statistical analysis seems inappropriate. Limes 63 and 64, the authors make a general assumption regarding the gene silencing induced by pathogens. This assumption is too general and it is not applied in many models.Regarding the methodology, the authors use a general acetyl lysine antibody in the experiments. They should have used a more precise mark for Histone acetylation in the assays. By no means, Figure 3 presents specific HDAC1 activity data. Did the have immunoprecipitated HDAC1 prior to the activity measurement?There is no reason for the use of NAB in the experiments. NaB is a pan HDAC inhibitor and the results are not really useful. Instead, they should have used another HADC1 specific inhibitor. The authors should have shown the level of transfection, the percentage of transfected cells when they proceeded with the si experiments. Figure 4 shows an important effect of scrambled si on the parasite load. Please, explain.Reviewer #2: 1-To elaborate more about the strain that has been used from Dr Dr StephenBeverly2-To add a picture related to the mechanism of action of siRNAReviewer #3: In general, the hypothesis is provocative, and the data are (on the surface) quite convinci

---

## [Editor Report · Decision Letter 1]

24 Feb 2020

Dear Dr. Muthuswami,

We are pleased to inform you that your manuscript 'Epigenetic Regulation of Defense Genes by Histone Deacetylase 1 in Human Cell Line-Derived Macrophages Promotes Intracellular Survival of Leishmania donovani.' has been provisionally accepted for publication in PLOS Neglected Tropical Diseases.

Before your manuscript can be formally accepted you will need to complete some formatting changes, which you will receive in a follow up email. A member of our team will be in touch within two working days with a set of requests.

Best regards,

Armando Jardim, PhD

Associate Editor

Hans-Peter Fuehrer

Deputy Editor

---

## [Editor Report · Acceptance letter]

25 Mar 2020

Dear Dr. Muthuswami,

We are delighted to inform you that your manuscript, "Epigenetic Regulation of Defense Genes by Histone Deacetylase 1 in Human Cell Line-Derived Macrophages Promotes Intracellular Survival of Leishmania donovani.," has been formally accepted for publication in PLOS Neglected Tropical Diseases.

Best regards,

Serap Aksoy

Editor-in-Chief

Shaden Kamhawi

Editor-in-Chief
